The morphology and internal structure of dogwood (Cornus L.) endocarps in the taxonomy and phylogeny of the genus

Morozowska Maria 1 maria.morozowska@up.poznan.pl
Woźnicka Agata 2
http://orcid.org/0000-0002-8947-9534 Nowińska Renata 1
1 Department of Botany, Faculty of Agronomy, Horticulture and Bioengineering, Poznań University of Life Sciences , Poznań , Poland
2 Legnica , Poland
Maloof Julin
Electronic publication date: 2021 Oct 28
Publication date: 2021
Volume: 9
Electronic Location ID: e12170
Received 2020 Nov 26; Accepted 2021 Aug 26
Copyright: © 2021 Morozowska et al.
Copyright year: 2021
Copyright holder: Morozowska et al.
License: This is an open access article distributed under the terms of the Creative Commons Attribution License, which permits unrestricted use, distribution, reproduction and adaptation in any medium and for any purpose provided that it is properly attributed. For attribution, the original author(s), title, publication source (PeerJ) and either DOI or URL of the article must be cited.
License URL: https://creativecommons.org/licenses/by/4.0/

Keywords: Fruit stones, Micromorphology, SEM, Sculpture, Subgenera, Species

Funding: Polish Ministry of Science and Higher Education Research Capacity Grant 506.641.01.00 and 507.641 Polish Ministry of Science and Higher Education’s program 005/RID/2018/19 This research was supported by the Polish Ministry of Science and Higher Education Research Capacity Grant for the Poznań University of Life Sciences (No. 506.141.01.00) and Grants to fund research of young scientists and participants in PhD studies (No 507.641). Publication was co-financed within the framework of the Polish Ministry of Science and Higher Education’s program: “Regional Initiative Excellence” in the years 2019–2022 (No. 005/RID/2018/19). The funders had no role in study design, data collection and analysis, decision to publish, or preparation of the manuscript.

==============================
The genus Cornus is known for its complicated taxonomy and extreme polymorphism. Currently, dogwoods are classified into four morphological groups reflected in four phylogenetic clades: blue- or white-fruited dogwoods (BW), cornelian cherries (CC), big-bracted dogwoods (BB), and dwarf dogwoods (DW). The present study is a continuation of a series of papers that started with the examination of the endocarp morphological diversity among BW species. The endocarps of 22 Cornus species were studied according to their morphology, internal structure, and sculpture; the aim was to evaluate the taxonomic importance of the examined characters and check whether endocarp differentiation supports the published phylogenies, according to which the relationships within the genus are indicated as (BW(CC(DW-BB))). Among the quantitative characters, the endocarp’s length, thickness, and width, its length/width ratio, and the number of vascular bundles on its surface were considered taxonomically important. Regarding the qualitative characters, the taxonomic significance of the stone shape, the endocarp apex and base outline, the position of vascular bundles on the endocarp surface, and the presence of the apical cavity and distinct furrow on the endocarp side walls was proved. Additionally, the uniform qualitative characters having a 100% share of a given character state were identified. Cornelian cherries and dwarf dogwoods were characterised by the presence of four uniform characters. In the big-bracted group, two uniform characters were found. Blue- or white-fruited species were the most heterogeneous, with no uniform characters. Regarding the endocarp’s internal structure, the presence of secretory cavities in the endocarp wall and in the septa, the presence of crystals in the outer endocarp, the number of cell layers in the transition sclereid zone, and the primary and secondary sculptures were found to be taxonomically significant. Additionally, the comparative analyses of dogwood endocarps found the following parameters to be useful: the germination valve thickness, the ratio of endocarp wall thickness to endocarp diameter, and the septum’s width and structure. Due to the great differentiation of the examined characters, it was difficult to verify the research hypothesis unambiguously. The calculated similarity coefficients of the endocarps’ qualitative and quantitative characters revealed the highest morphological similarity of endocarps between DW and BB groups, supporting the phylogenetic relationships based on molecular evidence. The number of vascular bundles on the endocarp surface was the character that supported this similarity the most. The similarity coefficients calculated separately for qualitative characters showed the closest similarity between groups CC-DW. However, these results do not directly reflect any of the published phylogenies.

Introduction

The monophyly of Cornus L. is strongly supported by morphological and molecular data (Murrell, 1993; Xiang et al., 1993, 1996; Xiang, Soltis & Soltis, 1998; Fan & Xiang, 2001, 2003). However, due to the genus’ extreme polymorphism, the relationships between different taxonomic units (subgenera, sections) into which the genus was divided in the past were unclear for almost a century (for details, see Xiang et al., 2006). The genus comprises approximately 60 species classified into 10 subgenera, which belong to four morphological groups: blue- or white-fruited dogwoods (BW) [Yinquania (Zhu) Murrell, Kraniopsis Raf., Mesomora Raf.], cornelian cherries (CC) [Afrocrania (Harms) Wangerin, Cornus L., Sinocornus Q. Y. Xiang], big-bracted dogwoods (BB) [Discocrania (Harms) Wangerin, Cynoxylon Raf., Syncarpea (Nakai) Xiang], and dwarf dogwoods (DW) (Arctocrania Endl. Ex Reichenbach). Similar to the morphological classification, four major clades—BW, CC, BB, and DW—have been identified within the genus, resulting from phylogenetic studies based on either morphological, molecular, or combined data (Xiang et al., 1996, 2006; Fan & Xiang, 2001, 2003). However, the relationships of the particular groups, subgenera, or species based on the morphological or molecular evidence were incongruent (Murrell, 1993; Xiang et al., 2006).

Murrell (1993), through his cladistic analysis of 28 morphological, anatomical, chemical, and cytological characters, performed a simultaneous analysis of relationships within the genus and generated a hypothesis on the Cornus phylogeny as (BW(DW(CC-BB))). Such hypothesis, in which the closest relationship was found between the CC and BB groups, differed to some extent from those proposed earlier (Adams, 1949; Bate-Smith et al., 1975; Jahnke, 1986; Eyde, 1988). However, this typology’s unique feature was the placement of the dwarf dogwoods with subgenus Arctocrania as the sister group basal to the clade formed by the subgenera Afrocrania, Cornus, Discocrania, Cynoxylon, and Syncarpea (see Figs. 1 and 2 in Xiang et al., 2006). Subsequently, Xiang, Soltis & Soltis (1998) and Fan & Xiang (2001) showed the phylogenetic relationships between four clades within genus Cornus as (BW(CC(DW-BB))). However, these relationships were supported by low bootstrap values, so they were not clearly resolved. According to Xiang et al. (2006), the differences between the molecular and morphological data arose because none of the previous molecular analyses included all ten subgenera of Cornus, and the samplings were limited. Furthermore, in Murrell’s (1993) cladistic analysis of morphological characters, the chosen outgroup taxa were not complete as only the genera Mastixia and Diplopanax were taken into consideration. Xiang et al. (2006) generated DNA sequences for maturase K (matK) and internal transcribed spacer (ITS) with a complete sampling of species to reconstruct a species-level phylogeny of the genus. The authors reconsidered Murrell’s (1993) morphological data matrix and expanded it by introducing nine new characters; they added the genus Alangium (sister of Cornus) as an outgroup. This provided improved resolution of the species’ relationships. However, the relationships between subgenera and clades were still different depending on the type of the data considered. The results of the phylogenetic analyses of the morphological data suggested that the BB and CC groups were sisters, but they were only moderately supported by the bootstrap values. According to the relationships suggested by the molecular evidence, the matK trees showed that DW-BB and CC-BW were two pairs of sister clades, with the latter being weakly supported (65%). In turn, the ITS trees placed the CC group with the BB-DW clade rather than with the BW group. The combined matK-ITS data set showed relationships similar to those of the matK and ITS trees but with higher support for most of the nodes. Complex analyses performed with the use of the combined results of phylogenetic and morphological studies, together with the already published rbcL and 26S rDNA sequences, resolved the relationships among clades as (BW(CC(DW-BB))) (Xiang et al., 2006).

The combined analyses of Xiang et al. (2006) included 37 morphological characters and all DNA sequences using parsimony and Bayesian methods. Most morphological characters included inflorescence and flower structure, leaves and locules, and the chromosome number and selected chemical compounds. The stony endocarps, the reproductive structures of potentially high taxonomic value, were examined only within a limited range. Only three characters of endocarps were analysed: shape, cavities, and apex of the fruit stone. Recently, extensive studies on dogwoods’ morphology, including their stony endocarps, were performed by Schulz (2011, 2012), Zieliński et al. (2014), and Woźnicka, Melosik & Morozowska (2015). Schulz (2011) gave an overview of the morphological differentiation within the genus, including the fruit stones’ morphology, and discussed the various affinity of the particular groups and subgroups considering the Cornus genus’ great polymorphism. In his next work, Schulz (2012) characterised the alternate-leafed dogwoods of the subgenera Mesomora and Kraniopsis (BW), including the cultivated taxa. The author stressed the importance of the endocarp morphology in the taxonomy of dogwoods and proposed a new taxonomic approach for some of the very closely related species, for example, C. alba L., C. sericea L. (= C. stolonifera Michx.), and C. occidentalis [(Torr. & A. Gray) Coville]; C. amomum (Mill.) and C. obliqua (Raf.); C. australis (C. A. Mey) and C. sanguinea (L.); C. foemina (Mill.) and C. racemosa (Lam.). Taking into consideration the earlier suggestions of Wangerin (1910), Schulz (2012) proposed the rank of the subspecies within Cornus alba as follows: C. alba L. subsp. alba and C. alba subsp. stolonifera (Michx.). Later, Zieliński et al. (2014) summarised and described the history of C. alba and C. sericea taxonomy. The authors discussed the problems with the identification of these very similar and closely related species and agreed that the broad species concept of C. alba s.l. (including C. sericea) was most reliable. Justifying such a position, Zieliński et al. (2014) also stressed that wild plants in generative and vegetative states could be properly identified and that cultivars of uncertain origin could have an easier and less controversial classification.

Woźnicka, Melosik & Morozowska (2015), who described the qualitative and quantitative differences in the morphology of endocarps of 15 Cornus species from the BW group, proved the taxonomic and systematic importance of the endocarp morphology. However, their study couldn’t fully explain species status in the case of closely related taxa, as some of the observed morphological differences overlapped or were too subtle. In the published dichotomous key based on the morphology of endocarps, Woźnicka, Melosik & Morozowska (2015) adopted the new taxonomic treatments of very closely related species proposed by Schulz (2012). The authors discussed the taxonomic importance and systematic implications of the obtained results in a phylogenetic framework. Further, a partial congruence between the observed morphological differentiation of endocarps and a currently available species phylogeny within the BW clade of the genus Cornus was found.

Apart from the modern dogwoods, the well-preserved fruits and woody endocarps of the extinct representatives of the Cornaceae family and the whole Cornales clade are by far the most taxonomically informative fossils, which facilitate a better understanding of the initial phylogenetic diversification of Cornales (Eyde, 1987, 1988; Manchester, Xiang & Xiang, 2010; Atkinson, Stockey & Rothwell, 2016; Stockey, Nishida & Atkinson, 2016; Atkinson, Stockey & Rothwell, 2017). The present study continues a series of papers that started with Woźnicka, Melosik & Morozowska (2015), who examined the diversity within endocarp morphology among selected species from the blue- and white-fruited dogwoods. The present work focused on the morphology of fruit stones of 22 species representing the whole genus Cornus; it used the same methods employed by Woźnicka, Melosik & Morozowska (2015). Additionally, our present approach was extended to analyse the endocarps’ sculpture and internal structure. The main aim of our study was to verify the taxonomical importance of the examined characters. We also compiled the earlier and presently obtained results to discuss our findings in a broader context, considering the phylogenetic relationships within the entire Cornus genus. We hypothesised that the grouping of the studied dogwoods based on the endocarps’ morphology and internal structure coincides with phylogenetic relationships (BW(CC(DW-BB))) based on the combined molecular and morphological data described by Xiang et al. (2006).

Materials & methods

Materials

Endocarps of 22 Cornus species representing four morphological groups (BW, CC, BB, and DW) were collected (Table 1). The selection of species depended only on their availability in botanical and herbarium collections. Initially, 27 species were selected for the study. Unfortunately, in the case of five species (C. volkensii Harms and C. chinensis Wangerin subg. Sinocornus, C. hongkongensis Hemsl subg. Syncarpea, C. oblonga Wall. subg. Yinquania, and C. disciflora Moc & Sessé ex DC. subg. Discocrania), the obtained material was insufficient, so they were not included in the study. For the sources of endocarps of 15 blue- or white-fruited dogwoods, see Woźnicka, Melosik & Morozowska (2015). The stones of red-fruited Cornus species were collected from cultivated plants growing in 10 Polish and 8 other European, Asian, or American botanical collections (Supplemental Files 1 and 2) and from 12 herbarium collections: BM, BG1, G, GH, H, K, KOR, KRAM, L, POZ, S, TRN (Thiers, 2021) (Supplemental File 3). Plant materials were collected between 2009 and 2012, from July to October, during the fruiting period of individual species. The endocarps were extracted from the fully developed ripened fruits.

Table 1 List of the examined Cornus species.

Species No	Cornus species	Species affinity	
Subgenus	Morphological group	
1	alba L.	Kraniopsis	BW	
2	alternifolia L.f.	Mesomora		
3	amomum Mill.	Kraniopsis		
4	australis C.A.Mey.	Kraniopsis		
5	bretschneideri L.Henry	Kraniopsis		
6	controversa Hemsl.	Mesomora		
7	drummondii C.A.Mey.	Kraniopsis		
8	foemina Mill.	Kraniopsis		
9	macrophylla Wall.	Kraniopsis		
10	obliqua Raf.	Kraniopsis		
11	occidentalis (Torr. & A.Gray) Coville	Kraniopsis		
12	racemosa Lam.	Kraniopsis		
13	sanguinea L.	Kraniopsis		
14	sericea L.	Kraniopsis		
15	walteri Wangerin	Kraniopsis		
16	mas L.	Cornus	CC	
17	officinalis Siebold & Zucc.	Cornus		
18	florida L.	Cynoxylon	BB	
19	kousa F.Buerger ex Hance	Syncarpea		
20	nuttallii Audubon ex Torr. & A.Gray	Cynoxylon		
21	canadensis L.	Arctocrania	DW	
22	suecica L.	Arctocrania		

The examination of the endocarp morphology was partly based on our earlier work; we included in our analyses the results obtained by Woźnicka, Melosik & Morozowska (2015), who examined 2,812 stones collected from 185 specimens representing 15 dogwood species of the blue- or white-fruited dogwoods from the BW group. Here we examined 1,034 stones collected from 69 specimens representing seven red-fruited Cornus species of the CC, BB, and DW groups. The combined data set of the endocarp morphology of 22 dogwood species representing all four groups (BW, CC, BB, and DW) was used for the complex analysis based on the results of morphological measurements of 3,846 stones collected from 254 specimens. Each species was represented by 3–15 specimens derived either from the herbarium or cultivated collections. Within these collected materials, 2–30 stones per specimen were evaluated depending on their availability. Details are provided in Supplemental File 4.

Within 22 species of dogwoods, the internal structures of 317 stones were examined. The number of tested endocarps per species depended on their availability. For most species, 15 endocarps each were tested. The exceptions were C. nuttallii and C. suecica, each with 10 endocarps tested, and C. canadensis, with 12 endocarps tested.

The original nomenclature of the examined dogwoods was considered appropriate and reliable. However, to increase the reliability of the study, approximately 70% of the specimens were verified according to their morphology (Supplemental Files 2 and 3). The nomenclature of the examined species follows The Plant List, version 1.1 (2013). The specimens and fruit materials were deposited in the herbarium of the Department of Botany (POZNB) at Poznań University of Life Sciences in Poland.

Plant measurements

Endocarp morphology

The seventeen morphological characters (eight qualitative and nine quantitative) of the woody stones were analysed (Table 2) for 3,846 stones collected. The terminology used to describe these 17 morphological characters of endocarps was from Woźnicka, Melosik & Morozowska (2015) (Table 2; Fig. 3). Since few of endocarps’ qualitative characters (ASH, apical shape; BSH, basal shape; SSH, shape in the vertical projection; and VBP, position of vascular bundles on the surface) were described in the different states, the percentage share of the particular states of the same character was determined for each species under study (Supplemental File 5). If the share of a given state was 100% of the occurrences, the character was regarded as uniform for the group, subgenera, or species. If the share of a character state was greater than or equal to 90%, it was regarded as typical for these taxa.

Figure 3 The mean (point), the standard deviation (box), and the minimum and maximum values (whisker) for the endocarp length (SL), thickness (ST), and width (SW).

(A, D, G) Groups; (B, E, H) subgenera; (C, F, I) species. Different lowercase letters indicate particular units differing in a given character (Dunnett’s T3 tests, p < 0.05). The red box marks the groups with the significant differences inside. For species description, see Table 1.

Table 2 The qualitative and quantitative characters examined in the morphometric analyses of Cornus endocarps.

No.	Description	Character
abbreviation	Type of character:
continuous (C)
discrete (D)	Units of
measurement/
coding	
1	Endocarp length	SL	C	mm	
2	Endocarp thickness	ST	C	mm	
3	Endocarp width	SW	C	mm	
4	Endocarp length-to-width ratio	SL/SW	C		
5	Endocarp width-to-thickness ratio	SW/ST	C		
6	Number of vascular bundles on endocarp surface	VN	C		
7	Share of forked vascular bundles	FV%	C	%	
8	Apical cavity length	ACL	C	mm	
9	Apical cavity width	ACW	C	mm	
10	Endocarp shape in vertical projection	SSH	D	0-2	
11	Apical shape	ASH	D	0-3	
12	Basal shape	BSH	D	0-3	
13	Smooth/rough endocarp surface	SSF	D	0-1	
14	Absence/presence of apical cavity	ACP	D	0-1	
15	Position of vascular bundles on endocarp surface	VBP	D	0-2	
16	Absence/presence of forked vascular bundles	FV	D	0-1	
17	Absence/presence of distinctive furrow	DF	D	0-1	

Endocarp cross-section

Four quantitative characters of the endocarp structure visible on the endocarp equatorial cross-section were analysed (Table 3, Fig. 1) for 317 stones. Longitudinal sections of the stones were also submitted for scanning electron microscopy (SEM). The endocarps were sectioned with a Leica CM18050 cryostat in a cryochamber at −15 °C. The ImageJ application (Rasband, 1997–2018) was used for measurements. A Zeiss Axioscope A1 stereoscopic microscope was used for photographic documentation.

Figure 1 Measurements of the characters of the endocarp internal structure: C. mas cross section (×6.9), GVT, SMW.

GVT, germination valve thickness; SMW, septum width.

Table 3 The quantitative characters of the internal structure of Cornus endocarps.

No.	Description	Character
abbreviation	Type of character:
continuous (C) discrete (D)	Units of
measurement	
1	Germination valve thickness	GVT	C	mm	
2	Septum width	SMW	C	mm	
3	Thickness of endocarp wall (germination valve) divided by endocarp diameter, multiplied by 100	WTP	C	%	
4	Number of cavities in endocarp wall	DCN	D		

Data analysis

Endocarp morphology

The comparative analyses of the results were made at three levels: groups (BW, CC, DW, and BB), subgenera (Kraniopsis, Mesomora, Cornus, Cynoxylon, Syncarpea, and Arctocrania), and species.

The statistical analyses were based on the average values of the measurements of 17 morphological characters obtained from 254 specimens considered in the present work. The quantitative characters of the stones were transformed logarithmically to obtain a normal or close to normal data distribution for statistical purposes (Sárnal et al., 1999; Howell, 2007; Tabachnick & Fidell, 2007).

The principal component analysis (PCA) aimed to estimate the diversity of specimens in terms of the quantitative morphological characters, identify the main trends in the diversity of the set of specimens, and select the quantitative characters that were most closely related to the observed gradient of diversity. The PCA was used for all quantitative characters, including apical cavity length (ACL) and width (ACW). The apical cavity was present in 12 out of 22 species and was a constant character only in three of them. Using PCA allowed us to determine whether such very strongly variable characters have a significant influence on the differentiation of the Cornus genus. If endocarps had no apical cavity, zero was entered for ACL and ACW. First, all data were standardised due to the wide range of values. The calculations were based on the characters’ correlation matrix and varimax rotation. The Kaiser criterion was used to select the principal components (V) that significantly explained the variability of the set. Therefore, the principal components whose eigenvalue exceeded or was close to 1.0 were left. The PCA enabled the identification of the characters strongly correlated (r ≥ 0.60) with previously selected principal components.

The multivariate analyses of variance (MANOVAs) based on the three PC scores were carried out to determine whether a separation of endocarps in terms of their belonging to groups, subgenera, and species was statistically significant. To evaluate this differentiation, each PC axis was compared using analyses of variance (ANOVAs) with the Bonferroni correction.

At the next stage of statistical analysis, the specimens from particular groups, subgenera, and species were assessed for significant differences in the morphological characters selected through the PCA method. The homogeneity of variances of characters was checked with Levene’s test. As some of the characters did not meet the assumption of the homogeneity of variance, the significance of differences was verified using one-way ANOVA with the Welch correction and Dunnett’s T3 post-hoc test.

The statistical analyses of the eight qualitative characters were based on individual measurements of 3,846 stones. Chi-square tests of independence were used to test the relationship between specific qualitative characters and the belonging of the dogwood specimens to particular groups, subgenera, and species. The Yates correction was used in the analyses with one degree of freedom (df = 1). With this correction, the discrete distribution of the characters was better approximated by the continuous chi-square distribution (Aczel, 2000). When the expected value was less than 5, the closest states of the characters were combined into one class (in tests with df > 1), or Fisher’s exact test was used (in tests with df = 1).

To check whether the studied morphological characters of the endocarps reflected the current relations within the Cornus genus (Murrell, 1993; Xiang et al., 2006), similarity coefficients were calculated. For eight log-transformed quantitative characters (one character was excluded during the course of analyses; see Results), the Manhattan distance converted into similarity was calculated. In the case of eight qualitative characters, the Jaccard similarity coefficient was used. The qualitative characters were common in several states, and the same character states were often observed in the four compared groups of dogwood. Therefore, the similarities between the groups concerned a similar frequency of occurrence of character states. For this reason, when calculating the Jaccard index, the number of endocarps sharing the same character states was considered as the intersection for each pair and was divided by the total remaining number of endocarps for that pair. For both quantitative and qualitative characters, Gower’s similarity coefficient was developed. The obtained similarity coefficients were in the range [0 1], with 1 denoting the highest similarity (the compared groups were identical) and 0 denoting the lowest.

Most statistical analyses were performed using the program STATISTICA 11 (StatSoft, Poland). ANOVAs with Welch’s correction and Dunnett’s T3 tests were calculated with IBM SPPS 21.0 (SPSS Inc., Chicago, IL, USA). The Bonferroni correction was found at http://quantitativeskills.com/sisa/calculations/bonfer.htm. The chi-square tests of independence with df > 1were performed at http://www.quantpsy.org/chisq/chisq.htm.

Endocarp cross-section

The discriminant function analysis (DFA) was used to check which of the species under study were the most strongly discriminated on the basis of three examined characters of the endocarps present in all groups studied (GVT, germination valve thickness; SMW, septum width; and WTP, thickness of the endocarp wall divided by the endocarp diameter, multiplied by 100). Factor structure coefficients, that is, simple correlations between the characters and discriminatory axes were used. Each character was tested statistically to verify the initial assessment of diversity obtained through the DFA. Some of the data did not have a normal distribution after the logarithmic transformation either. Therefore, nonparametric tests were used on nontransformed data. The Kruskal–Wallis H test was used for three combinations of categories (four groups, six subgenera, 22 species). Numerous Mann–Whitney U tests were also conducted to compare consecutive pairs within each category. Statistical analyses were performed using STATISTICA 11 (StatSoft, Poland).

SEM analysis

Endocarp cross-sections and longitudinal sections were made at half of the endocarps’ lengths or widths, respectively. Whole endocarps and their cuttings were gold-coated and examined with a Zeiss EVO 40 electron microscope. The following surface micromorphological characters were examined: cellular pattern, cell outline (tetragonal, polygonal, rounded, irregular), anticlinal walls (straight, sinuate), relief of the anticlinal cell boundary (raised, channelled), and curvature of the outer periclinal cell wall (concave, convex, flat). According to endocarps’ internal structure, the following characters were examined: the outer endocarp (OE) (absent, present), the type of sclereids in the OE (isodiametric, elongated), crystals in the OE (absent, present), the inner endocarp (IE) (absent, present), the transitional sclereid zone (TS) (absent, present/the number of cell layers), the structure of the endocarp septum (S) (solid, openwork, partly openwork), and crystals in S (absent, present).

Results

Endocarp morphology

Quantitative characters

The diversity of the quantitative morphological characters was initially tested using PCA. The first three principal components together explained 77.95% of the total variance (V1 = 35.08%; V2 = 31.98%; V3 = 14.74%). The scatter diagram of the first two components showed a clear cluster of specimens belonging to the same species and those belonging to the same subgenus and group (Fig. 2). The groups and subgenera were clearly separated, with the exception of the CC group and the Cynoxylon subgenus from the BB group, which overlapped. The first principal component mostly separated the DW and CC groups (Table 4). Four characters were negatively correlated with the first principal component: the endocarp thickness (ST), the apical cavity width (ACW), the apical cavity length (ACL), and the endocarp width (SW) (Table 5). The second component separates the above-mentioned groups from the species included in the BW group. The following characters were negatively correlated with the second component: the endocarp length (SL) and the endocarp length-to-width ratio (SL/SW). The following characters were positively correlated: the share of bifurcated vascular bundles (FV%) and the number of vascular bundles on the endocarp surface (VN). The third principal component noticeably separated the BB and DW groups (Table 4). The following characters were positively correlated: the apical cavity width (ACW) and the apical cavity length (ACL). The endocarp width-to-thickness ratio (SW/ST) was not significantly correlated with any of the principal components (Table 5). Therefore, this character was not included in further analyses.

Figure 2 A scatterplot of two PCA components (V1, V2) for nine quantitative characters of the endocarps based on the mean values of 254 examined Cornus specimens.

BW, CC, BB, DW–groups; Mesomora, Kraniopsis, Syncarpea, Cynoxylon–subgenera.

Table 4 The average factor loadings of the groups in the PCA for a sample of 254 Cornus specimens.

Groups	Axis 1	Axis 2	Axis 3	
DW	3.46	−1.10	1.98	
BW	−0.21	0.91	−0.19	
CC	−2.06	−3.57	−0.06	
BB	0.25	−2.65	−0.31	

Table 5 The PCA results for nine quantitative characters of the Cornus endocarps representing 22 species from four groups.

PCA axis	V 1	V 2	V 3	
Eigenvalues:	3.16	2.88	1.33	
Character	Component loadings	
SL	−0.57	−0.70	−0.20	
ST	−0.83	−0.24	−0.42	
SW	−0.83	0.09	−0.41	
SL/SW	−0.10	−0.90	0.06	
SW/ST	0.20	0.53	0.10	
VN	−0.24	0.85	−0.29	
FV%	−0.46	0.69	−0.17	
ACL	−0.76	0.11	0.64	
ACW	−0.75	0.15	0.64	
Total variance explain. %	35.08	31.98	14.74	
Cumul. total variance explain. %	35.08	66.25	77.95	
Note:

Strong correlations (r ≥ 0.6) marked in bold. For a description of the characters, see Table 2.

MANOVAs based on PC scores (V1–V3), with the taxonomic affiliations as independent variables, confirmed that tested groups, subgenera, and species differed in the quantitative characters of endocarps (groups: Wilks’ lambda = 0.03, F(3,9) = 203.67, P < 0.0001; subgenera: Wilks’ lambda = 0.00, F(3,15) = 311.6, P < 0.0001; species: Wilks’ lambda = 0.00, F(3,63) = 113.05, P < 0.0001). The statistically significant separation within taxonomic units occurred along all tree axes (ANOVAs for groups: F(3,250) = 69.37, 555.46, and 32.94, respectively, P < 0.0001; ANOVAs for subgenera: F(5,248) = 262.58, 479.63, and 122.72, P < 0.0001; ANOVAs for species: F(21,232) = 150.51, 164.95, and 51.62, P < 0.0001).

The differentiation of the particular quantitative characters of the endocarps between individual groups, subgenera, and species as shown in Figs. 3–5. The endocarp length (SL) and thickness (ST) were the best diagnostic characters as they significantly differentiated each group from the others (Figs. 3A and 3D; post-hoc tests, p < 0.05). The length and thickness of the endocarps (SL and ST) were the greatest in the CC group and the smallest in the DW group. The endocarp width (SW) and length-to-width ratio (SL/SW) also assumed the highest values in the CC group and the smallest values in the DW group. However, these two characters did not differ significantly between the BW and BB groups (Figs. 3G and 4A). The highest number of vascular bundles (VN) was present on endocarps of species from the BW group. This character distinguished all groups, with the exception of BB and DW (Fig. 4D).

Figure 4 The mean (point), the standard deviation (box), and the minimum and maximum values (whisker) for the endocarp SL/SW ratio, and the number of vascular bundles on the endocarp surface (VN).

SL/SW–endocarp length-to-width ratio. (A, D) Groups; (B, E) subgenera; (C, F) species. Different lowercase letters indicate particular units differing in a given character (Dunnett’s T3 tests, p < 0.05). The red box marks the groups with the significant differences inside. For species description, see Table 1.

Figure 5 The mean (point), the standard deviation (box), and the minimum and maximum values (whisker) for the apical cavity length (ACL) and apical cavity width (ACW) in three Cornus species.

Different lowercase letters indicate particular units differing in a given character (Dunnett’s T3 tests, p < 0.05). The red box marks the groups with the significant differences inside. For species description, see Table 1.

Considering the differentiation in quantitative characters within particular subgenera, the endocarp length (SL) also revealed a significant difference between subg. Mesomora and subg. Kraniopsis as well as between subg. Cynoxylon and subg. Syncarpea (Fig. 3B). Subgenera Mesomora and Kraniopsis also differed significantly according to endocarp thickness (ST) and width (SW) (Figs. 3E and 3H). The endocarp length/width ratio (SL/SW) significantly differed between subg. Cynoxylon and subg. Syncarpea (Fig. 4B). The number of vascular bundles (VN) did not differ between subgenera Mesomora and Kraniopsis, between Cornus and Cynoxylon, and between Cynoxylon, Syncarpea, and Arctocrania (Fig. 4E).

Most of the species clustered in the same groups and subgenera exhibited great similarities in the analysed quantitative characters, but with some exceptions. The endocarp length (SL) and SL/SW ratio of C. kousa (Syncarpea) were significantly lower compared to those of C. florida and C. nuttallii (Cynoxylon) (Figs. 3C and 4C; species nos. 18–20). The endocarp thickness (ST) of C. alba, C. bretschneideri, C. occidentalis, and C. sericea were significantly smaller than those of other BW species (Fig. 3F; species nos. 1, 5, 11, and 14). The endocarp width (SW) of C. canadensis was significantly smaller than that of C. suecica (Fig. 3I; species nos. 21–22). Some similarities between species clustered in different subgenera were also observed. C. kousa (Synacrpea) and both C. canadensis and C. suecica from subg. Arctocrania did not differ (Dunnett’s T3 test, p > 0.05) in endocarp length (SL) and SL/SW ratio (Figs. 3B, 3C, 4B, and 4C). In turn, the endocarp thickness (ST) of Cornus officinalis (Dunnett’s T3 test, p > 0.05) did not differ from those of species from subg. Cynoxylon and Syncarpea (Figs. 3E and 3F).

An apical cavity was observed in stones of 12 out of 22 species under study. However, for most of these species, it occurred sporadically (C. australis, C. bretschneideri, C. drummondii, C. foemina, C. macrophylla, C. obliqua, C. sanguinea, C. walteri, and C. officinalis). Thus, the apical cavity length and width (ACL, ACW) were analysed in detail only for C. alternifolia, C. controversa (Mesomora), and C. mas. Both species of subg. Mesomora differed from each other in ACL and ACW. Furthermore, their cavities were significantly longer and wider comparing to C. mas (Figs. 5A and 5B).

Qualitative characters

The results of the chi-square test of independence showed significant differences between examined dogwood groups, subgenera, and species in all the qualitative morphological characters of the endocarps (Table 6).

Table 6 The results of the chi-square test of independence between the qualitative characters of the Cornus endocarps and the affinity of the specimens to groups, subgenera, and species, N = 3,846.

Character	Group	Subgenus	Species	
df	Chi2	p	df	Chi2	p	df	Chi2	p	
SSH	9	2405.31	0.00	15	5209.32	0.00	63	8567.18	0.00	
ASH	9	1652.95	0.00	15	1921.87	0.00	63	5072.13	0.00	
BSH	9	361.07	0.00	15	837.07	0.00	63	2493.70	0.00	
SSF	3	260.74	0.00	5	763.87	0.00	21	2868.97	0.00	
ACP	3	474.17	0.00	5	2166.57	0.00	21	2706.49	0.00	
VBP	6	1001.28	0.00	10	3196.48	0.00	42	6852.60	0.00	
FV	3	1641.09	0.00	5	1714.26	0.00	21	2190.12	0.00	
DF	3	881.36	0.00	30	1048.30	0.00	21	2101.29	0.00	
Note:

For a description of the characters, see Table 2.

None of the examined characters were uniform within all groups, subgenera, or species. However, it was possible to distinguish uniform characters for particular groups, subgenera, and species (Table 7).

Table 7 The list of uniform qualitative characters of Cormus endocarps within groups, subgenera, and species.

Character	Group	Subgenus		Species		
SSH	CC – spherical	Cornus – spherical	CC	C. walteri – spherical	BW	
		Cynoxylon – spherical	BB	C. mas – spherical	CC	
		Syncarpea – irregular		C. officinalis –spherical		
				C. florida – spherical	BB	
				C. kousa – irregular		
				C. nuttalii – spherical		
				C. canadensis – spherical	DW	
				C. suecica - flattened		
ASH	CC – rounded or truncate	Mesomora – rounded or truncate	BW	C. alternifolia – rounded or truncate	BW	
		Cornus – rounded or truncate	CC	C. controversa – rounded or truncate		
				C. macrophylla – rounded or truncate		
				C. mas – rounded or truncate	CC	
				C. officinalis – rounded or truncate		
				C. canadensis – acuminate	DW	
BSH	-	-		C. macrophylla – rounded	BW	
				C. officinalis – rounded	CC	
				C. canadensis – rounded	DW	
SSF	CC – smooth	Cornus – smooth	CC	C. alba – smooth	BW	
	DW - smooth	Cynoxylon – smooth	BB	C. alternifolia – smooth		
		Arctocrania – smooth	DW	C. bretschneideri – smooth		
				C. drummondii – smooth		
				C. foemina – smooth		
				C. macrophylla – smooth		
				C. obliqua – rough		
				C. occidentalis – smooth		
				C. racemosa – smooth		
				C. sericea – smooth	BW	
				C. walteri – smooth		
				C. mas – smooth	CC	
				C. officinalis – smooth		
				C. florida – smooth	BB	
				C. nuttalii – smooth		
				C. canadensis – smooth	DW	
				C. suecica - smooth		
VBP	DW – flat	Mesomora – sunken	BW	C. alternifolia – sunken	BW	
		Syncarpea – sunken	BB	C. amomum – raised		
		Arctocrania – flat	DW	C. bretschneideri – flat		
				C. controversa – sunken		
				C. drummondii – flat		
				C. foemina – flat		
				C. macrophylla – flat		
				C. obliqua – raised		
				C. occidentalis – flat		
				C. sanguinea – flat		
				C. sericea – flat	BW	
				C. walteri – flat		
				C. officinalis – flat	CC	
				C. kousa – sunken	BB	
				C. nuttalii – flat		
				C. canadensis – flat	DW	
				C. suecica – flat		
FV	BB – unforked	Mesomora – forked	BW	C. controversa – forked	BW	
	DW – unforked	Cynoxylon – unforked	BB	C. mas – unforked	CC	
		Arctocrania – unforked	DW	C. florida – unforked	BB	
				C. canadensis – unforked	DW	
				C. suecica - unforked		
DF	CC – absence	Cornus – absence	CC	C. alternifolia – absence	BW	
				C. foemina – absence	BW	
				C. racemosa – absence		
				C. walteri – absence		
				C. mas – absence	CC	
				C. officinalis – absence		
				C. florida – presence	BB	
				C. suecica – absence	DW	

The qualitative characters were the least differentiated in the DW group, where four characters of endocarps were uniform: smooth surface (SSF), absence of apical cavity (ACP), flat vascular bundles (VBP), and unforked vascular bundles (FV). Additionally, two typical characters (>90% of the occurrences) were identified in DW: rounded base (BSH) and absence of distinctive furrow (DF), with 92.2% and 96.8% percentage of occurrence, respectively. In the CC group, four characters were considered uniform: spherical = globose endocarps (SSH), rounded or truncated apex (ASH), smooth endocarp surface (SSF) and absence of distinctive furrow (DF). In the BB group, two uniform characters were observed: absence of the apical cavity (ACP) and unforked vascular bundles (FV), while rounded or truncate apex (ASH) was recognised as a typical character, with a percentage of occurrence of 91.1%. The endocarps of BW species were the most heterogeneous, and neither uniform nor typical characters in this group were present.

Four characters were found to be uniform in the following subgenera: Mesomora [rounded or truncate apex (ASH), presence of apical cavity (ACP), sunken vascular bundles (VBP) and forked vascular bundles (FV)]; Cornus [spherical=globose endocarps (SSH), rounded or truncate apex (ASH), smooth endocarp surface (SSF), absence of distinctive furrow (DF)]; Arctocrania [smooth endocarp surface (SSF), absence of apical cavity (ACP), flat vascular bundles (VBP), unforked vascular bundles (FV)]. Three uniform characters were found in two subgenera: Cynoxylon [spherical=globose endocarps (SSH), smooth endocarp surface (SSF), unforked vascular bundles (FV)]; Syncarpea [irregular shape (SSH), absence of apical cavity (ACP) and sunken vascular bundles (VBP)] while in subg. Kraniopsis no uniform characters were present.

With reference to uniform characters recognised in particular species, the smooth endocarp surface (SSF) was observed in endocarps of 16 species (Figs. 6–7). In turn, flat vascular bundles (VBP) were present in endocarps of 12 species, the absence of distinctive furrow (DF) was uniform for seven species, a spherical=globose endocarp (SSH) was uniform for six species, a rounded or truncate apex (ASH) was uniform for five species, unforked vascular bundles (FV) were uniform for four species, sunken vascular bundles (VBP) and a rounded basal shape for the endocarp (BSH) were uniform in three species, and raised vascular bundles (VBP) were uniform for two species (Figs. 6–7, Table 7, Supplemental File 5).

Figure 6 The endocarp shape (SSH), apical shape (ASH), basal shape (BSH), and endocarp surface sculpture (SSF).

(A, D, G, J) Groups; (B, E, H, K) subgenera; (C, F, I, L) species. Different lowercase letters indicate particular units differing in a given character (chi-square tests, p < 0.05). The red box marks the groups with the significant differences inside (for species description, see Table 1). SSH: 0–spherical = globose; 1–intermediate; 2–flattened = compressed; 3–irregular. ASH: 0–shortly acuminate; 1–acuminate; 2–wedge-shaped; 3–rounded or truncate. BSH: 0–shortly acuminate; 1–long acuminate; 2–rounded; 3–wedge-shaped. SSF: 0–smooth; 1–rough.

Figure 7 The presence of the apical cavity (ACP), the vascular bundle position on the endocarp surface (VBP), the presence of bifurcated vascular bundles (FV), and the presence of a distinctive furrow (DF) on Cornus endocarps.

(A, D, G, J) Groups; (B, E, H, K) subgenera; (C, F, I, L) species. Different lowercase letters indicate particular units differing in a given character (chi-square tests, p < 0.05). The red box marks the groups with the significant differences inside (for species description, see Table 1). ACP: 0–without cavity; 1–with cavity. VBP: 0–sunken; 1–flat; 2–raised. FV: 0–unforked; 1–forked. DF: 0–absence; 1–presence.

Considering the obtained results based on the analyses of the uniform characters, no differences were found between the CC and DW groups in terms of smooth endocarp surface (SSF) and between the BB and DW groups in terms of unforked vascular bundles (FV). With reference to subgenera, no differences were found between Cornus and Cynoxylon in terms of spherical endocarps (SSH); between Mesomora and Cornus in terms of rounded or truncate apex (ASH); between Cornus, Cynoxylon, and Arctocrania in terms of smooth endocarp surface (SSF); between Mesomora and Syncarpea in terms of sunken vascular bundles (VBP); and between Cynoxylon, Syncarpea, and Arctocrania in terms of unforked vacular bundles (FV). The results concerning the occurrence of uniform characters in individual species are presented in Table 7.

The analysis results based on the examination of all endocarp characters concerning the differentiation between the studied groups, the subgenera and species showed that the endocarp shape (SSH), the apical shape (ASH), the basal shape of the endocarp (BSH), the position of vascular bundles on the endocarp surface (VBP), and the absence/presence of distinctive furrow (DF) enabled the differentiation of each group from the others (chi-square tests, p < 0.05) (Figs. 6A, 6D, 6G, 7D, and 7J). The absence/presence of apical cavity (ACP), and the absence/presence of forked vascular bundles (FV) separated the examined groups from each other, with the exception of BB and DW (Figs. 7A and 7G). The endocarp surface (SSF) enabled the differentiation of all the groups except the CC and DW (Fig. 6J).

None of the characters under study differentiated between all subgenera. However, the rounded or truncate apex (ASH) significantly distinguished between subg. Cornus and subg. Cynoxylon and Syncarpea, while subg. Cornus and Mesomora did not differ according to ASH (Fig. 6E). The rounded basal shape (BSH) was present in 68.7–92.2% of endocarps of subg. Kraniopsis, Mesomora, Cornus, Syncarpea, and Arctocrania. Subgenus Cornus did not differ between subg. Kraniopsis and Mesomora in terms of BSH, while the differentiation between subg. Syncarpea and Arctocrania was significant despite the observed similarity (Fig. 6H). The smooth endocarp surface (SSF) was predominant among subgenera Kraniopsis, Mesomora, Cornus, Cynoxylon, and Arctorania (78.3–100% of the occurrences within these subgenera), with the only exception of subg. Syncarpea; no differences were found between subg. Kraniopsis-Mesomora and Cornus-Cynoxylon-Arctorania. In subg. Syncarpea (C. kousa), 86.3% of endocarps had rough surfaces; thus, this subgenus differed significantly from all others in terms of SSF (Fig. 6K). The rough endocarp surface (SSF) was recognised also in 95.1% and 100% of endocarps of two Kraniopsis species: C. amomum and C. obliqua, respectively (Fig. 6L). The sunken vascular bundles (VBP) did not differ between subg. Mesomora and Syncarpea, while significant differences in VBP were found between all other subgenera (Fig. 7E). With reference to the presence of unforked vascular bundles (FV), there were no differences between subg. Cynoxylon, Syncarpea, and Arctocrania (Fig. 7H). The significant differentiation in FV was found within the subg. Cornus as unforked vascular bundles were present on 100% of Cornus mas and on 42.2% of C. officinalis endocarps. In turn, the unforked vascular bundles were present on 99.5% of C. kousa endocarps (Syncarpea) (Fig. 7I, Supplemental File 5). On endocarps of Kraniopsis and Mesomora, the forked vascular bundles were present in 77.7% and 99.8%, respectively (Fig. 7H). The absence of distinctive furrow (DF) was predominant on endocarps of subg. Mesomora (96.6%) and Arctocrania (96.8%), and no differences were found between these two subgenera. Same as above, the following pairs of subgenera: Mesomora-Cornus (96.6–100%) and Cornus-Arctocrania (100–96.8%) were very similar in terms of the absence of DF. However, they still had differences. The presence of DF was predominant among the endocarps of the Cynoxylon and Syncarpea specimens (90.7% and 86.3%), and no differences between these two subgenera were found (Fig. 7K).

Qualitative and quantitative characters

Similarity coefficients calculated for the morphological characters of the endocarps showed a high similarity between the CC, BB, and DW groups (Table 8). In all comparisons, the BW group had the lowest similarity to other groups. In terms of quantitative characters, the BB-DW groups were clearly the most similar (similarity coefficient S = 0.71). In the case of qualitative characters, the CC-DW groups were the most similar (S = 0.45), but the groups CC-BB were only slightly less similar (S = 0.42). Considering both quantitative and qualitative characters, the BB-DW groups were again most similar (S = 0.54).

Table 8 Similarity coefficients calculated for Cornus groups based on quantitative and qualitative characters of endocarps.

	Quantitative characters (N = 8)	Qualitative characters (N = 8)	All morphological characters (N = 16)	
	BW	CC	BB	DW	BW	CC	BB	DW	BW	CC	BB	DW	
BW	1.00				1.00				1.00				
CC	0.18	1.00			0.16	1.00			0.17	1.00			
BB	0.16	0.46	1.00		0.15	0.42	1.00		0.15	0.44	1.00		
DW	0.00	0.17	0.71	1.00	0.08	0.45	0.36	1.00	0.04	0.31	0.54	1.00	
Note:

The highest coefficients are bolded.

Endocarp cross-section

The discriminant analysis showed that the characters of internal structure of endocarps clearly distinguished the species of the CC group from other dogwoods and that the BW group overlapped with the DW and BB groups (Fig. 8). All the examined characters had significant discriminatory power (Table 9). There was a strong positive correlation between the valve thickness (GVT) and the first discriminant function. Similarly, a positive but weaker correlation was found for the next two characters: the septum width (SMW) and the thickness of the endocarp wall divided by the diameter of the endocarp, multiplied by 100 (WTP). The first discriminant function distinguished C. mas from other species and the DW group from the BB group. The second discriminant function separated C. officinalis. The WTP ratio was strongly and negatively correlated with this function. The other two characters (GVT and SMW) were less strongly correlated (Table 9).

Figure 8 A scatterplot of the first two discriminant functions (CAN1, CAN2) for 22 Cornus species based on the characters of the internal structure of 317 endocarps.

Table 9 Canonical discriminant function (CDA), the discriminatory power and correlations with the discriminant functions of three characters of the internal structure of Cornus endocarps.

Character	Wilk’s lambda	Partial lambda	Factor structure coefficients	
CAN1	CAN2	
GVT	0.12	0.39	0.84	−0.54	
SMW	0.10	0.46	0.50	0.55	
WTP	0.10	0.48	0.41	−0.79	
Eigenvalues	–	–	3.2	2.24	
Total variance explain. %	–	–	53.69	37.27	
Cumul. total variance explain. %	–	–	53.69	90.96	
Note:

For a description of the characters, see Table 3.

The Kruskal–Wallis tests (H) confirmed that GVT, SMW, and WTP were significantly different within each analysed level: group, subgenus, and species (Table 10). The valve thickness (GVT) and the WTP ratio significantly differentiated the stones of specimens from all four group (Figs. 9A and 9G). The highest values of both characters were noted in the CC, whereas the lowest were in the DW. The third character, that is, the septum width (SMW), enabled only the distinction between the BW and BB from DW (Fig. 9D).

Figure 9 The median (point), the first and third quartile (box) and the minimum and maximum values (whisker) for the germination valve thickness (GVT), septum width (SMW) and WTP in Cornus species.

WTP—thickness of the endocarp wall divided by the diameter of the endocarp, multiplied by 100. (A, D, G) groups; (B, E, H) subgenera; (C, F, I) species. Different lowercase letters indicate particular units differing in given characters. The red box marks the groups with significant differences inside (Mann–Whitney U tests, p < 0.05). For species description, see Table 1.

Table 10 The results of the Kruskal–Wallis tests (H) between the characters of internal structure of dogwood endocarps assigned to the group, subgenus, and species.

Taxonomic unit	Character	H	df	p	
Group	GVT	143.82		0.00	
	SMW	22.42	3	0.00	
	WTP	97.91		0.00	
Subgenus	GVT	145.60		0.00	
	SMW	64.02	5	0.00	
	WTP	102.43		0.00	
Species	GVT	231.27		0.00	
	SMW	218.48	21	0.00	
	WTP	178.59		0.00	
Note:

For a description of the characters, see Table 3.

The subgenera from the BW group differed significantly according to the examined cross-section characters. When comparing the endocarps of Mesomora and Kraniopsis (Figs. 9B, 9E, and 9H), the valve thickness (GVT) and the septum width (SMW) were significantly higher in the former, while the WTP ratio was significantly higher in the latter. In the BB group, the Cynoxylon species had stones with higher values of septum width (SMW) than Syncarpea representatives (Fig. 9E).

Among the particular species, the examined characters differentiated clearly between C. mas and C. officinalis in terms of the presence and absence of the septum (it was absent in C. officinalis) and the WTP ratio, which was significantly higher in C. officinalis (Figs. 9F and 9I; species nos. 16 and 17). The above-mentioned character differed between C. canadensis and C. suecica (Fig. 9I; species nos. 21 and 22).

Secretory cavities occurred only on the endocarp wall of C. mas and C. officinalis fruit stones. The average number of cavities (DCN) was significantly higher in C. mas endocarps (Mann–Whitney U test: U = 9.5, Z = 4.15, p = 0.000; Fig. 10).

Figure 10 The median (point), the first and third quartile (box), and the minimum and maximum values (whisker) for the number of cavities in the endocarp wall (DCN) in C. mas and C. officinalis.

The lowercase letters indicate significant differences (Mann–Whitney U tests, p < 0.05)

SEM analysis

The endocarps of most of the examined dogwoods were composed of the isodiametric sclereids with evenly thickened and lignified cell walls with numerous pits. In the structure of the endocarps of C. walteri from the BW group and those of C. florida, C. nuttallii, and C. kousa from the BB group, the elongated sclereids were also observed (Table 11, Figs. 11G, 12I, 12K, and 12M).

Figure 11 SEM micrographs of sections of endocarps of C. racemosa (A–B), C. macrophylla (C–D), C. foemina (E–F), C. walteri (G–H), C. amomum (I–J), C. obliqua (K–L), C. alternifolia (M–N), C. controversa (O–P).

(A, B) Cross-sections of endocarp and septum with crystals; (C, D) longitudinal sections of endocarp and septum with crystal; (E) longitudinal section of endocarp; (F) cross-section of septum with crystals; (G) cross-section of endocarp; (H) longitudinal section of septum with crystal; (I) cross-section of endocarp; (J) longitudinal section of septum with crystal; (K, L) cross-sections of endocarp with crystal and septum; (M) cross-section of endocarp with crystal; (N) longitudinal section of septum; (O, P) cross-sections of endocarp and septum with crystal. C, crystals; E, endosperm; Em, embryo; S, septum; TS, transition sclereid zone; IE, inner endocarp.

Figure 12 SEM micrographs of cross-sections and longitudinal sections of endocarps of C. mas (A–B), C. officinalis (C–D), C. suecica (E–F), C. canadensis (G–H), C. florida (I–J), C. nuttallii (K–L), C. kousa (M–N).

(A) Cross-section of endocarp; (B) cross-section of septum, endosperm, and embryo; (C, D) longitudinal sections of endocarp with endosperm, embryo, and crystals; (E) cross-section of endocarp; (F) longitudinal section of septum; (G, H) cross-sections of endocarp and septum with crystal; (I) cross-sections and longitudinal sections of endocarp with crystals; (J) cross-section of septum with crystals; (K) cross-sections and longitudinal sections of endocarp with crystal; (L) longitudinal section of septum with crystals; (M) cross-section and longitudinal sections of endocarp with crystal; (N) cross-section of septum with crystals. C, crystals; E, endosperm; Em, embryo; IE, inner endocarp; OE, outer endocarp; S, septum; TS, transition sclereid zone.

Table 11 The internal structure characters of the endocarps of 22 Cornus species under study.

Species	Outer endocarp (OE)	Inner endocarp (IE)	Transition sclereids (S)	Septum structure (S)	
sclereids	crystals	absent	present	absent	present (number of cell layers)	solid	openwork	partly openwork	crystals	
isodiametric	elongated	
BW	
C. alba	+			+			+ (2-3)			+	+	
A. alternifolia	+		+	+			+ (3-4)			+	+	
C. amomum	+			+			+ (3-4)		+		+	
C. australis	+		+	+			+ (3-4)			+	+	
C. bretchsneideri	+			+			+ (4-5)			+	+	
C. controversa	+			+			+ (1-1)	+			+	
C. drummondii	+		+	+			+ (5-6)	+			+	
C. foemina	+			+			+ (2-3)			+	+	
C. macrophylla	+			+			+ (2-3)	+			+	
C. obliqua	+			+			+ (3-4)		+		+	
C. occidentalis	+			+			+ (2-3)		+		+	
C. racemosa	+			+			+ (5-7)	+			+	
C. sanguinea	+			+			+ (2-4)			+	+	
C. sericea	+		+		+		+ (3-5)			+		
C. walteri	+	+			+		+ (1-2)			+	+	
CC	
C. mas	+	+			+	+		+*				
C. officinalis	+	+			+		+ (1-1)					
BB	
C. florida	+	+		+			+ (2-3)		+		+	
C. kousa	+	+		+		+				+	+	
C. nuttallii	+	+		+			+ (1-1)	+			+	
DW	
C. canadensis	+			+		+				+	+	
C. suecica	+			+		+				+		
Note:

* The septum with a secretory cavity.

In the endocarps of the species from the CC group (C. mas and C. officinalis), the outer (OE) and inner endocarps (IE) were distinguished. The OE was composed mostly of isodiametric sclereids and elongated fibrous cells surrounding the secretory cavities. The IE, interpreted also as the internal epidermis system (sensu Kaniewski & Hausbrandt, 1968), was composed of approximately six layers of parallel fibres with evenly thickened cell walls and numerous pits; it was surrounding the seed chambers (Figs. 12A and 12C).

It was difficult to distinguish the inner endocarp (IE) in most of the other species from the BB, BW, and DW groups. However, in endocarps of two species, C. walteri and C. sericea, the groups of cells with slightly thinner cell walls, compared to the whole endocarp, were observed to surround the seed chambers (Figs. 11G and 13D). Additionally, in the OE of four BW species (C. alternifolia, C. sericea, C. drummondii, and C. australis), the crystals were observed (Table 11, Figs. 11M, 13E, 13K, and 13U).

Figure 13 SEM micrographs of cross- and longitudinal sections of endocarps of C. alba (A–C), C. sericea (D–F), C. occidentalis (G–I), C. drummondii (J–L), C. bretchsneideri (M–O), C. sanguinea (P–R), C. australis (S–U).

(A) Cross-sections of endocarp and septum with crystal (B, C); (D, E) cross-sections of endocarp with crystal; (F) longitudinal section of septum; (G) cross-section of endocarp; (H, I) longitudinal sections of septum with crystal; (J, K) cross-sections of endocarp with crystals and septum (L); (M, N) cross-sections of endocarp and septum with endosperm and embryo; (O) longitudinal section of septum with crystal; (P) cross-sections of endocarp and septum with crystals (Q, R); (S) cross-sections of endocarp and septum with endosperm and crystals (T, U). C, crystals; E, endosperm; Em, embryo; S, septum; TS, transition sclereid zone; IE, inner endocarp.

Immediately under the surface of the endocarp, the transitional sclereid zone (TS) was present in most of the species. It was composed of 5(3)-1 layers of thin-walled cells (compared with the rest of the endocarp). These cells were slightly elongated and flattened in the plane parallel to the stone surface (Table 11, Figs. 11A, 11C, 11I, 11M, 12D, 12I, 13A, 13D, 13E, 13G, 13J, 13M, and 13P).

Isodiametric and elongated sclereids with differently thickened cell walls also appeared in the septa, which was present in the stones of all examined dogwoods, except C. officinalis. The differences in the structure of septa, depending on the degree of the cell wall thickness, were observed on the cross-sections and longitudinal sections of the stones. Three types of the septa were distinguished according to the cell wall thickness: (1) solid septa built in the cells with strongly thickened cell walls (Figs. 11B, 11D, 11P, 12B, 12L, and 13L), (2) partly openwork septa built in the outer part of the cells with strongly thickened cell walls and in the middle of the cells with thinner cell walls (Figs. 11F, 11H, 11N, 12F, 12H, 12N, 13B, 13F, 13N, 13Q, and 13T), and (3) openwork septa entirely composed of cells with slightly thickened walls and with large cell lumina (Figs. 11J, 11L, 12J, and 13H). In most of the species from the BW, BB, and DW groups, crystals were observed in the septa (Table 11; Figs. 11B, 11D, 11F, 11H, 11J, 11P, 12H, 12J, 12L, 12N, 13C, 13I, 13O, 13R, and 13U). A little different septa structure was observed in C. mas, a representative of the CC group. The septa were built from narrow strongly elongated fibres. Additionally, small secretory cavities were present in the septa (Fig. 12B).

The endocarp surface micro-ornamentation pattern was reticulate for most of the species under study. Surface cells were similar in size for particular species, mostly quadrangular to polygonal, sometimes rounded or irregular in their outline (Figs. 14–16). For some species like C. sericea, C. sanguinea, or C. amomum, the cellular pattern was less distinct (Figs. 14D, 14N, and 15J). The biggest cells were observed on C. mas, C. officinalis, and C. canadensis endocarps (Figs. 16B, 16D, and 16H). The anticlinal cell walls were straight and raised for most of the species but were wavy on C. racemosa and C. controversa endocarps, the two species from different subgenera: Kraniopsis and Mesomora, respectively (Figs. 15B and 15P). In C. alba, C. occidentalis, C. drummondii, C. bretschneideri, C. racemosa, and C. foemina, the specific constrictions and spherical thickenings were present on the anticlinal cell walls (Figs. 14B, 14F, 14J, 14L, 15B, and 15F). The outer periclinal cell walls were flat or concave, most often without the secondary sculpture (Table 12). The exceptions were the endocarps with the verrucose (C. sericea, C. occidentalis, and C. drummondii), striate (C. sanguinea and C. amomum), punctate (C. australis), or foveate (C. kousa) secondary micro-ornamentation pattern (Figs. 14D, 14H, 14J, 14N, 14P, 15J, and 16N).

Figure 14 SEM micrographs of endocarp sculpture of Cornus alba (A–B), C. sericea (C–D), C. occidentalis (E–H), C. drummondii (I–J), C. bretschneideri (K–L), C. sanguinea (M–N), C. australis (O–P).

Figure 15 SEM micrographs of endocarp sculpture of Cornus racemosa (A–B), C. macrophylla (C–D), C. foemina (E–F), C. walteri (G–H), C. amomum (I–J), C. obliqua (K–L), C. alternifolia (M–N), C. controversa (O–P).

Figure 16 SEM micrographs of endocarp sculpture of Cornus mas (A–B), C. officinalis (C–D), C. suecica (E–F), C. canadensis (G–H), C. florida (I–J), C. nuttallii (K–L), C. kousa (M–N).

Table 12 The micromorphology of the endocarp surface of 22 Cornus species under study.

Species	Cellular pattern	Cell outline	Anticlinal cell wall boundary	Outer periclinal cel wall	
tetragonal	polygonal	rounded	irregular	raised	straight	undulate	concave	flat	Secondary sculpture	
BW	
C. alba	reticulate		+			+	+			+		
A. alternifolia	reticulate		+			+	+			+		
C. amomum	reticulate		+	+		+				+	striate	
C. australis	reticulate		+	+		+	+		+		punctate	
C. bretchsneideri	reticulate		+			+	+		+			
C. controversa	reticulate				+	+		+		+		
C. drummondii	reticulate		+			+	+			+	verrucose	
C. foemina	reticulate		+			+	+			+		
C. macrophylla	reticulate		+			+	+			+		
C. obliqua	reticulate		+			+	+		+			
C. occidentalis	reticulate		+			+	+		+		verrucose	
C. racemosa	reticulate				+	+		+	+			
C. sanguinea	reticulate		+	+		+	+		+		striate	
C. sericea	reticulate		+	+		+	+		+		verrucose	
C. walteri	reticulate		+	+		+	+			+		
CC	
C. mas	reticulate		+	+		+	+		+			
C. officinalis	reticulate		+	+		+	+		+			
BB	
C. florida	reticulate	+				+	+			+		
C. kousa	reticulate	+		+		+	+			+	foveate	
C. nuttallii	reticulate	+				+	+			+		
DW	
C. canadensis	reticulate	+	+			+	+		+			
C. suecica	reticulate	+	+			+	+		+			

Discussion

The present research provides documentation of the great diversity of the endocarp characters and indicates the usefulness of some of them in the identification and systematic assessment of many Cornus species. The results are related to recent studies on taxonomy (Schulz, 2011, 2012; Zieliński et al., 2014; Woźnicka, Melosik & Morozowska, 2015), and they refer to published phylogenies of the genus Cornus (Murrell, 1993; Xiang et al., 2006).

Endocarp morphology

Quantitative characters

Considering the taxonomic importance of the examined quantitative characters, the endocarp length (SL) and its thickness (ST) were identified as the best diagnostic characters, as they significantly differentiated each group from others. The number of vascular bundles (VN) was the character that significantly differentiated almost all BW species from the red-fruited species (CC, BB, and DW). It was shown that the specimens representing the Kraniopsis-Mesomora complex (BW) had significantly more vascular bundles than the specimens representing the subgenera Cornus, Cynoxylon, Syncarpea, and Arctocrania (Figs. 4E and 4F). Within the red-fruited dogwoods, the average number of vascular bundles (VN) differed significantly for the CC, BB, and DW species, with the exception of C. nuttallii (Cynoxylon, BB) (Figs. 4D, 4E, and 4F). Such results agree with phylogenies based on molecular evidence (Xiang et al., 2006). However, the similarity of C. mas and C. officinalis (subg. Cornus, CC) with C. nuttallii in terms of VN supports Murrell’s (1993) morphological phylogeny.

The results on the subgeneric level showed that there were no differences in the endocarp length (SL) between Syncarpea and Arctocrania species. Additionally, the same character significantly differentiated between subg. Cornus and subgenera Cynoxylon, Syncarpea as well Arctocrania (Fig. 3B). Exactly the same similarities/differences were found according to the SL/SW ratio (Fig. 4B). The differentiation of SL and SL/SW found among the above-mentioned subgenera indicates that the described results support the molecular (Xiang et al., 2006) rather than the morphological (Murrell, 1993) phylogenies.

Considering other studied quantitative characters, such as the endocarp width (SW) and its thickness (ST), the results were not so clear. The SW analyses showed that subgenera Cornus and Mesomora did not differ in that character, which may highlight the results of the parsimony analysis of matK, according to which CC-BW was the pair of sister clades (Xiang et al., 2006). In turn, the stone thickness (ST) did not differ significantly between subg. Cornus and subg. Cynoxylon and Syncarpea, which supports Murrell’s (1993) phylogeny.

Qualitative characters

Among the qualitative characters, the endocarp shape in vertical projection (SSH), apical shape (ASH), basal shape (BSH), smooth/rough endocarp surface (SSF), position of vascular bundles on endocarp surface (VBP), absence/presence of forked vascular bundles (FV), and absence/presence of distinct furrow (DF) were taxonomically important characters, as they allowed us to differentiate between particular species, subgenera, or groups.

The differentiation in the stone shape (SSH) allowed the selection of the species with spherical, intermediate, flattened, and irregular stones. Flattened fruit stones were quite often present for species from BW (33.7%) and DW (42.9%) groups, while spherical endocarps were uniform for the CC group. Irregular stones were uniform for Syncarpea and allowed a clear distinction of C. kousa from the rest of the examined dogwoods. The stone shape (SSH) together with the degree of flattening of the stone, i.e. its thickness (ST) and length-to-width ratio (SL/SW) were often used in the past to analyse the diversity of the endocarps of very similar and phylogenetically closely related species (e.g. C. alba, C. sericea, and C. occidentalis from the BW group) (Fosberg, 1942; Szafer & Pawłowski, 1959; Rehder, 1967; Bean, 1976; Seneta, 1994; Rutkowski, 2004; Xiang et al., 2006; Schulz, 2012; Zieliński et al., 2014; Woźnicka, Melosik & Morozowska, 2015). However, the results of recent studies concerning dogwoods’ morphology (Schulz, 2012; Zieliński et al., 2014; Woźnicka, Melosik & Morozowska, 2015) and our analyses of ST and SL/SW average values have shown that the ranges of these characters mostly overlap; thus, they do not explain sufficiently the species status of these closely related taxa. Additionally, the results concerning the flattened shape (SSH) of C. alba, C. sericea, and C. occidentalis endocarps (97.8%, 92.1%, and 94.4%, respectively) and the presence of distinctive furrow running longitudinally on the lateral faces of their endocarps (DF) (53.8%, 77.2%, and 87.9%, respectively) also indicated a close similarity between C. alba, C. sericea, and C. occidentalis (BW). Considering these three species, Schulz (2012) proposed the new taxonomic approach of C. alba s.l. with subsp. alba and subsp. stolonifera. He treated C. occidentalis at the rank of variety below subspecies stolonifera of C. alba. This new taxonomic approach was accepted by Zieliński et al. (2014) and Woźnicka, Melosik & Morozowska (2015) and supported by our results.

The lack of significant differentiation in the endocarp apical shape (ASH) and its basal shape (BSH) supports close relationships between particular species from the subg. Kraniopsis (BW) (e.g., North American species C. amomum, C. obliqua, C. foemina, and C. racemosa or European species C. sanguinea and C. australis). With reference to C. amomum and C. obliqua, some other morphological similarities between these two species [e.g. the rough endocarp surface (SSF) (95.1% and 100%, respectively) and the uniform raised vascular bundles (VBP)] were also found. Referring to C. sanguinea and C. australis, the flat vascular bundles (VBP) with 93.7% and 100% of the occurrences and unforked vascular bundles (FV) with over 91.5% of the occurrences indicated their close similarity. All the described results support Schulz (2012), who suggested that C. obliqua and C. australis should be classified as subspecies within C. amomum and C. sanguinea, respectively. The results also complement the earlier findings describing the morphological and molecular similarities of these species (Wilson, 1964; Bean, 1976; Ball, 2005; Xiang et al., 2006; Schulz, 2012; Woźnicka, Melosik & Morozowska, 2015).

The noticeable variation in the form of endocarp apical shape (ASH) allowed the distinction between examined Arctocrania species (DW). In turn, the rounded or truncated apex was recognised as a uniform character for C. mas and C. officinalis (CC). According to Bojnanský & Fargašová (2007), C. mas stones have a pointed apex, whereas C. officinalis stones have a blunt apex. It is most likely that the differences between the results of our study and the cited data were caused by the inverse description of the stone apex and the stone base by Bojnanský & Fargašová (2007). The rounded or truncated apex (ASH) was also recognised as a uniform character for alternate-leaf dogwoods C. alternifolia and C. controversa (subg. Mesomora). However, it should be noted that this character was somewhat combined with the presence of distinct ACP, which was present on the endocarps of these two species and was recognised as a taxonomically important character. It allowed for a significant separation of two alternate-leaf dogwoods C. alternifolia and C. controversa (subgen. Mesomora, BW) from species belonging to subg. Kraniopsis, which was in line with earlier results by other authors (Eyde, 1988; Xiang & Boufford, 2005; Schulz, 2011, 2012; Woźnicka, Melosik & Morozowska, 2015). Our results confirmed also the presence of noticeable, but clearly much smaller, apical cavities on the stones of several Kraniopsis species, which was described earlier by Woźnicka, Melosik & Morozowska (2015). Apart from the BW group, much smaller apical cavities were present on the endocarps of examined species from subg. Cornus. They were frequent on C. mas (99.9%) fruit stones and occurred sporadically on C. officinalis (16.1%) endocarps. We also observed a shallow depression on C. volkensii stones (unpublished data). Eyde (1988) did not clearly state whether cavities could be found in these species nor in what form. As far as the species from the group of large-fruited edible dogwoods (CC) were concerned, the author only described the apical cavity that was wide and shallow in C. volkensii; in C. chinensis, it had the form of a V-shaped incision and was absent on C. sessilis endocarps. According to Manchester, Xiang & Xiang (2010), the endocarps of species belonging to subg. Cornus differ in the expression of an apical cavity. These authors described a V-shaped apical notch in C. chinensis and a broad shallow cavity in C. volkensii. The other species were described as apically rounded without or with inconspicuous apical cavity. Atkinson, Stockey & Rothwell (2016) confirmed the presence of the apical cavity on stones of C. mas, C. officinalis, and C. volkensii, together with a few other living (C. chinensis, C. eydeana, and C. sessilis) and extinct species (C. piggae, C. ettingshausenii, and C. multilocularis). According to the results, the ACP in C. mas fruit stones was significantly smaller than in endocarps of C. alternifolia and C. controversa (Fig. 5). Based on the described results, the presence or absence of apical cavities is a taxonomically important character. However, according to already published data, it is a plesiomorphic character and thus have no value in the analysis of the phylogeny of the genus (Murrell, 1996).

The extensive and detailed study of Eyde (1988) also includes an analysis of the variation in a stone’s surface. The author emphasised that the stones of the herbaceous species (DW) were easy to distinguish because they were smoother than the stones in the BW group. Similar results were obtained earlier by Woźnicka, Melosik & Morozowska (2015), which were confirmed in the present study. The smooth endocarp surface (SSF) was found in all three groups of the red-fruited dogwoods (CC, BB, and DW), but with the exception of C. kousa (Syncarpea, BB) (Fig. 6L), which agrees with Eyde (1988, Figs. 13h and 13i, p. 276) who showed that C. kousa stones were irregular and rough. A uniform character of smooth endocarps present in three subgenera from different groups—Cornus (CC), Cynoxylon (BB), and Arctocrania (DW)—supports published phylogenies based on morphological and molecular data (Murrell, 1993; Xiang et al., 2006).

The vascular bundle position on the endocarp surface (VBP) differentiated significantly Kraniopsis and Mesomora species. The sunken vascular bundles were recognised as a uniform and taxonomically important character for subg. Mesomora. In endocarps of Kraniopsis species, mostly flat vascular bundles were present. The present results confirmed the results of Woźnicka, Melosik & Morozowska (2015). The significant differentiation in VBP within the subgenera of red-fruited dogwoods (CC, BB, and DW) also indicate systematic importance of that character. The presence of flat vascular bundles, which were uniform for the endocarps of a few species belonging to different morphological groups, like C. officinalis (CC), C. nuttallii (Cynoxylon, BB), C. canadensis, and C. suecica (DW), supports the phylogenies based on the morphological and molecular evidence (Murrell, 1993; Xiang et al., 2006). In turn, the presence of the unforked vascular bundles on endocarp surface (FV) showed a distinct similarity between two groups of red-fruited dogwoods (BB-DW), which supports the close phylogenetic relationship of these clades shown by Xiang et al. (2006). However, there was a close similarity between C. mas (subg. Cornus) and BB species according to the presence of the unforked vascular bundles, which supports Murrell’s (1993) morphological phylogeny.

The presence of distinctive furrow running longitudinally on the lateral faces of an endocarp (DF) was a strongly variable character. It was uniform only for the CC group, so no support for the published phylogenies was found according to that character. However, some of its taxonomic usefulness should be mentioned. The DF was present on 35.6% of Kraniopsis endocarps and only on 0.4% of Mesomora fruit stones (BW). Such results agree with the results of Fosberg (1942), Schulz (2012), and Woźnicka, Melosik & Morozowska (2015). In reference to the red-fruited dogwoods, fruit stones with furrows on sides were predominant among the C. florida, C. nuttallii, and C. kousa endocarps (BB) (86.3%–90.7%), while it was almost (96.8%) or completely (100.0%) absent on endocarps of CC and DW species. Such results indicate that the presence of DF on the endocarps of most of big-bracted species may be helpful in distinguishing them from other red-fruited dogwoods.

The calculated similarity coefficients allowed us to check whether the described morphological similarities between the studied groups and subgenera reflect their current phylogenetic relations. The similarity coefficients (Table 8) calculated jointly for the qualitative and quantitative characters showed that the endocarps of big-bracted dogwoods (BB) and dwarf dogwoods (DW) were the most similar. Such results indicated support for molecular phylogenies (Xiang et al., 2006). When only qualitative characters were considered, the closest similarity of groups CC-DW was found mainly due to the smooth endocarp surface (SSF). This character may possibly be considered a synapomorphy for CC and DW groups. However, that similarity does not directly reflect any of the published phylogenies (Murrell, 1993; Fan & Xiang, 2001; Xiang et al., 2006).

Endocarp cross-section

The results concerning the internal structure of endocarps of C. mas, C. officinalis, and C. volkensii (unpublished data) and the available literature data (Kaniewski & Hausbrandt, 1968; Eyde, 1988; Manchester, Xiang & Xiang, 2010) indicate that the numerous secretory cavities in both the stone walls and the septa are uniform and may be identified as synapomorphy for subg. Cornus. According to Manchester, Xiang & Xiang (2010), this character is taxonomically important as it allows the correct identification and classification of fossil endocarps of these species. Recently, Atkinson, Stockey & Rothwell (2016) examined the anatomy of permineralised fruits from the Campanian of Vancouver Island and showed that the presence of secretory cavities in the woody endocarps indicated that they were assignable to the Cornus subg. Cornus. Manchester, Xiang & Xiang (2010) also analysed the germination valve thickness (GVT) and the thickness of germination valve divided by endocarp diameter (WTP). The author found these parameters very useful in comparative analyses of endocarps of living dogwoods and fossil materials representing the former taxa of the Cornus subg. Cornus (CC). The same two characters were also included in our research, and the results were comparable to C. mas and C. officinalis fruit stones. In C. mas endocarps, the WTP was 21 and the GVT was 1.2 mm. In C. officinalis endocarps, the WTP ratio was 28 and the GVT was 1.1 mm. According to Manchester, Xiang & Xiang (2010), the analogous data were 20 and 0.8–1.2 mm for C. mas stones and 22–34 and 0.9 mm for C. officinalis endocarps.

The present research confirmed that the endocarp tissue of CC and BB species consists of isodiametric and elongated sclereids. Takahashi, Crane & Manchester (2003) named them “rise grain shaped sclereids.” Interestingly, the elongated sclereids were also present in the endocarps of C. walteri (BW), which means that this character was shared by two “unrelated” clades. To resolve whether it was a possible homoplasy, cladistic analyses were needed. With reference to the CC group, it was confirmed that the outer and inner regions of endocarp were clearly distinguished (Manchester, Xiang & Xiang, 2010; Morozowska, Gawrońska & Woźnicka, 2013; Atkinson, Stockey & Rothwell, 2016; Stockey, Nishida & Atkinson, 2016; Atkinson, Stockey & Rothwell, 2017).

The present study also described several new findings potentially helpful in the taxonomic examination of living and fossil woody fruit stones of different Cornus species. Endocarps of species belonging to subgenera Kraniopsis and Mesomora (BW) and of species from subgenera Syncarpea and Cynoxylon (BB) differed significantly in terms of GVT, SMW, and WTP (Figs. 9B, 9E, and 9H, respectively). Among red-fruited dogwoods, the highest values of GVT and WTP were typical for endocarps of CC species (Figs. 9A and 9G), while the narrowest septum (SMW) was typical for endocarps of DW species (Figs. 9D–9F), which correlated with their lowest dimensions (SL, ST, and SW) (Figs. 3A–3I).

SEM analyses

The comparisons between the internal structure and the micromorphology of endocarps of closely related species also highlighted some significant differences of taxonomic importance. Cornus alternifolia and C. controversa (BW, subg. Kraniopsis and Mesomora, respectively) endocarps differed according to such characters as the presence or absence of crystals in the outer endocarp (OE), the number of cell layers in the transition sclereids zone (TS), the septa structure (S), the cell outline, and the anticlinal walls of the surface cells (Tables 11 and 12). The presence of the undulate anticlinal cell walls differed significantly between C. controversa and C. alternifolia. Considering other phylogenetically, closely related species, such as C. alba–C. sericea–C. occidentalis, C. amomum–C. obliqua, C. foemina–C. racemosa–C. drummondii, or C. australis–C. sanguinea (Xiang et al., 2006), the results obtained showed that some of them were very similar, while the others clearly differed in terms of the internal structure and micromorphology of endocarps (Tables 11 and 12). The primary sculpture of the first three species was almost the same. However, C. sericea and C. occidentalis endocarps stood out with the presence of the verrucose secondary sculpture. Moreover, Cornus sericea was the most different species as far as endocarp’s internal structure was concerned. The differences included the presence of crystals in the OE, the absence of crystals in the septa, the presence of the inner endocarp (IE), and the multilayered zone of transition sclereids. The presence of IE was generally a rare character in BW species. Besides C. sericea, it was observed only in C. walterii stones. According to Kaniewski & Hausbrandt (1968), the inner endocarp is probably destroyed during the ripening of the dogwood fruit. With reference to C. amomum and C. obliqua, no differences were found in the internal structure of their endocarps, supporting the close phylogenetic relationship of these species, as shown by Xiang et al. (2006). However, it should be noted that the presence of the secondary sculpture on C. amomum endocarps was different compared to C. obliqua, which may be of taxonomic importance. Referring to C. foemina, C. racemosa, and C. drummondii, two of them (C. foemina and C. racemosa) were treated, together with C. macrocarpa Nash (= C. asperifolia Michaux) from eastern North America, as three subspecies of C. foemina complex due to their morphological similarity (all are white-fruited dogwoods) and many interspecific intermediate forms (Wilson, 1964). According to the results, some differences between C. foemina, C. racemosa, and C. drummondii were found. They differed in the number of cell layers forming the transition sclereids zone, the type of septa, and the form of anticlinal and periclinal cell walls (Tables 11 and 12). It should be stressed that the undulate anticlinal cell walls present on C. racemosa (subg. Kraniopsis) endocarps were exceptional among the examined dogwoods. Another species with the same type of anticlinal walls was C. controversa (subg. Mesomora). It turned out that this character was present on endocarps of two species with different origins belonging to two different subgenera. C. racemosa is the North American species with white fruits, while C. controversa is an Asian species with purplish red or bluish black fruits (Xiang & Boufford, 2005). Regarding C. australis and C. sanguinea, the results showed an almost complete resemblance in the internal structure and micromorphology of their endocarps. The only difference concerned the presence of crystals in the OE of C. australis fruit stones (Table 11). Such findings support the close phylogenetic relationship of these species described by Xiang et al. (2006).

Referring to red-fruited dogwoods (C. mas and C. officinalis), the cellular pattern on their endocarps was characterised by the largest surface cells, and their size and the endocarp size were correlated. Similarly, large surface cells were found on the endocarps of C. canadensis, but no similar correlation was found. In endocarps of the red-fruited dogwoods, as in BW species, no crystals were observed in OE. However, their presence was found in the stones of C. volkensii, subg. Afrocrania (unpublished data). To confirm the taxonomic significance of this character, the internal structure of the endocarps of all CC species must be examined.

Summing up the morphological differentiation of endocarps is largely consistent with the published phylogenies within the Cornus genus. Consistency with phylogenetic relations according to Xiang et al. (2006) is evident for quantitative characters and for combined quantitative and qualitative characters when the BB-DW groups are the closest to each other. In the case of qualitative characters, the relationships between groups are much more complex, as they show almost the same similarity between CC-DW and CC-BB, showing consistency with Murrell’s (1993) phylogeny. However, if only the uniform characters are compared, it turns out that the CC-BB groups do not have any common character, while the BB-DW groups have one common character (lack of forked vascular bundles on the endocarp surface). The results clearly indicate how complicated the Cornus genus is in terms of taxonomy and phylogeny.

Conclusions

Our results showed that despite the large diversity in endocarps’ morphological and internal structures, selected characters may be used to a greater or lesser extent in the taxonomy of the genus Cornus for both living and fossil materials. Fruits are by far the most taxonomically informative fossils in Cornales (Eyde, 1987, 1988; Atkinson, Stockey & Rothwell, 2016). They are used in analyses of phylogenetic relationships within Cornus and in examinations concerning the timing of the initial diversification and evolution of the family Cornaceae and the basal asterid lineage, Cornales (Atkinson, Stockey & Rothwell, 2016). We have demonstrated that the morphology and internal structure of endocarps allow for the differentiation of particular living species. In some cases, only one character was sufficient to distinguish one taxon from others, such as irregular stones found only in C. kousa (subg. Syncarpea, BB) or seven to eight vascular bundles running along the entire perimeter of the stones of BW species, and might probably be considered synapomorphies for these lineages. However, additional studies with the use of cladistic methods should be conducted in this case. Some specimens were differentiated based on sets of characters (e.g., much longer than wider endocarps with clearly visible valves and often with small apical cavities, which were characteristics of the Cornus subg). The micromorphology and internal structure of endocarps were also found to be potentially helpful in the taxonomy of dogwoods. The microstructure of the Cornus endocarp surface has not been studied previously. Earlier studies included the micromorphological structure of leaves of the selected dogwood species to assess the degree of diversity of the characters under analysis and to verify their taxonomic and phylogenetic usefulness (Hardin & Murrell, 1997; Schulz, 2012; Zieliński et al., 2014; Gawrońska et al., 2019). Since one of the limitations of the current study is the sample size, further replication studies on larger samples of the studied Cornus subgenera and species are recommended to confirm the described results.

Supplemental Information

Supplemental Information 1 List of the botanical collections from which the cultivated materials were collected.

Click here for additional data file.

Supplemental Information 2 The origin of the cultivated materials and their taxonomic verification.

Click here for additional data file.

Supplemental Information 3 The provenance and the geographic origin of the Cornus species herbarium accessions tested.

Click here for additional data file.

Supplemental Information 4 Number of tested Cornus endocarps per specimens, species, subgenera, and groups.

Click here for additional data file.

Supplemental Information 5 The number and percentage of endocarps of Cornus species, subgenera, and groups with various states of qualitative characters.

Click here for additional data file.

Supplemental Information 6 Raw data.

Click here for additional data file.

We would like to thank Mrs. Ilona Wysakowska from the Department of Botany for her technical assistance in preparing the manuscript.

Additional Information and Declarations

Competing Interests

Author Contributions

Data Availability

The authors declare that they have no competing interests.

Maria Morozowska conceived and designed the experiments, analyzed the data, prepared figures and/or tables, authored or reviewed drafts of the paper, and approved the final draft.

Agata Woźnicka conceived and designed the experiments, performed the experiments, analyzed the data, prepared figures and/or tables, authored or reviewed drafts of the paper, and approved the final draft.

Renata Nowińska conceived and designed the experiments, performed the experiments, analyzed the data, prepared figures and/or tables, authored or reviewed drafts of the paper, and approved the final draft.

The following information was supplied regarding data availability:

The raw data are available in the Supplemental File.

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
