# Peer review of "The morphology and internal structure of dogwood (Cornus L.) endocarps in the taxonomy and phylogeny of the genus"

_PeerJ, doi:10.7717/peerj.12170_

## Round 0.1 · original submission · Major Revisions

Please read carefully though the constructive comments made by the two reviewers. There is a need to present your data more clearly and to offer a more objective presentation of the degree to which characters fit the phylogeny.

Reviewer 1 ·

Basic reporting

This manuscript provides excellent documentation of the variation in features useful in the identification and systematic assessment of many species of Cornus fruits and gives good support for the value of certain suites of endocarp characters for taxonomic determinations. It will be useful to paleobotanists as well as systematists and is a good example of the kinds of analyses that would be usefully applied to other clades of angiosperms.

In the annotated copy of the manuscript I made minor corrections to English and a few critical comments. I consider the manuscript acceptable following minor revision. Here are some specific comments.

I was confused by the use of the term “anatomy” to refer to features usually thought of as morphology. Line 40: “With regard to the anatomy of endocarps, the most important features were the germinating valve thickness, the ratio of the endocarp wall thickness divided by the endocarp diameter, the septum width and its structure, the presence or absence of crystals in the outer endocarp, the number of cell layers in the transition sclereids zone and the primary and secondary sculpture”

Some of the characters listed here are traditionally considered morphology rather than anatomy: germination valve thickness, ratio of the endocarp wall thickness divided by the endocarp diameter, the septum width and its structure. Only the presence or absence of crystals in the outer endocarp, the number of cell layers in the transition sclereids zones are characters of anatomy/histology.

In my opinion, the acronyms (such as GVT, SMW and WTP) are unnecessary in the main text, but useful only in the figures and captions. The article will be much easier to read if the actual words are used. But if you do want to use them, each acronym should be explained at its first occurrence in the text (this has not been done consistently in this manuscript).

Experimental design

It may be appropriate in the material and methods to indicate which species of Cornus were not available or not studied. Although all major clades were sampled, not all of the species were presented.

Validity of the findings

Line 420: “ the presence of secretory cavities in the endocarp wall was a
species-specific trait. It occurred only in C. mas and C. officinalis fruit stones.”

I am curious why C. sessilis is not mentioned as another example. Were the cavities missing in your collections of C. sessilis? Figure 2 of Manchester et al 2010 shows cross sectioned stones with cavities present in C. mas, C. officianalis, C. sessilis, C. volkensia, C. eydeana, C. chinensis and this is thought to be a feature of all the species of subgenus Cornus (Cornelian cherries), so it might be a subgenus trait rather than species-specific trait.

Annotated reviews are not available for download in order to protect the identity of reviewers who chose to remain anonymous.

·

Basic reporting

In general, the manuscript contains valuable and interesting descriptions and analysis of Cornus endocarps. There are problems with the grammar and sentence structure and I’ve tried to make some corrections. The data set is complex. The tables do not maintain character lists in any order, so it is difficult to compare among tables and text. The characters are not well described in some places and there are problems with calling characters and character states “variables”, “variants” and “traits”. References are appropriate. Structure is problematic. There is a need for better organization to help the reader keep track of the characters. Table 6 description states that “(for trait description [sic] see Table 5)”, yet there are no trait descriptions in Table 5. Maybe Table 2? Discussion of traits does not follow in order of Tables 9 and 10. This would help the reader understand the results and how they inform our understanding of the phylogeny. Figure 8. A dendrogram is presented, yet there was no discussion in the manuscript of how this was generated. What data were used?

Experimental design

The manuscript is evaluating phylogenetic studies and asking if the presented data supports or does not support the phylogenies, yet the data are not presented in a character and character state way that allows the reader to evaluate the value of the character for phylogenetic analysis. Many of the character states are variable, yet those that vary within taxa are used interchangeably with those that do not. It would be useful to differentiate those characters or character states that are invariant, recognize these as synapomorphies, and then ask the question of whether these synapomorphies support the published phylogenies.

Validity of the findings

The statement “based on the morphological features of their endocarps exactly reflect their phylogenetic relations BW(CC(DW-BB)) shown by Fan & Xiang (2001) and Xiang et al. (2006) was verified positively” is not supported by the results presented. It appears that data are picked for discussion that support certain phylogenies, but those that do not support the published phylogenies are ignored. For example, Figure 2 shows overlap of CC and BB, but this is ignored in the discussion. In another example, Figure 9 circumscribes the BB and DW groups, but ignores the overlap with the BW taxa with DW and BB, but not with CC. No outgroups were examined. Traits such as lack of apical cavity are pleisomorphic, so have no value in analysis of the phylogeny of the genus. Traits were described as supporting particular phylogenies when the trait was not uniform within the taxa. Many of the character states described are variable within taxon, yet the phylogenetic utility of the described characters is not discussed.

Additional comments

I have made comments on the manuscript.

---

## Round 0.2 · Minor Revisions

I am sorry for the delay in getting this back to you. The previous academic editor is no longer available and it took some time for me to take over. There are some minor edits suggested by the reviewer that need to be addressed before publication (see annotated PDF). I am OK if you leave the final sentence of the intro as is (since it is a hypothesis). I suspect you can make these changes quickly and I will expedite approval as soon as I get it back.

Reviewer 1 ·

Basic reporting

This article is much improved following the previous round of review.

Experimental design

no further comment following earlier rerview

Validity of the findings

Yes, well presented.

Additional comments

I have accepted your rebuttal to some of my earlier recommendations. In the annotated pdf, I have indicated a few minor corrections an places for clarification.

Annotated reviews are not available for download in order to protect the identity of reviewers who chose to remain anonymous.

---

## Round 0.3 · accepted · Accept

Thank you for addressing the final round of reviewer comments. I am recommending this for publication.